# Mapping Tribo-Corrosion Behaviour of TI-6AL-4V Eli in Laboratory Simulated Hip Joint Environments

**Ghulam Rasool** *, **Yousuf El Shafei** and **Margaret M. Stack** 

Department of Mechanical and Aerospace Engineering, University of Strathclyde-James Weir Building, 75 Montrose Street, Glasgow G1 1XJ, UK; yousuf.el-shafei.2015@strath.ac.uk (Y.E.S.); margaret.stack@strath.ac.uk (M.M.S.)
\* Correspondence: ghulam.rasool@strath.ac.uk

**Abstract:** Wear and corrosion in artificial hip replacements are known to result in metal ion release and wear debris induced osteolysis. This may lead to pain and sensitivity for patients. This infers that pre-clinical testing is critical in determining the long-term performance, safety, and reliability of the implant materials. For this purpose, micro-abrasion-corrosion tests were carried out on a biocompatible material, Ti-6Al-4V ELI, using a T-66, Plint micro-abrasion test rig in conjunction with Gill Ac corrosion testing apparatus for the range of applied loads and electrical potentials in the hip joint simulated environment. A Ringer's solution, with and without an abrasive particle (silicon carbide), was used to enable the interactions between abrasion and corrosion. In this paper, the effects of applied load and electrochemical potential on the tribo-corrosion behaviour of Ti-6Al-4V in a bio-simulated environment are presented. The wastage, micro-abrasion-corrosion mechanisms, and synergy behaviour were identified and mapped. A significant difference in corrosion current densities was observed in the presence of abrasive particles, suggesting the removal of the protective oxide layer. The results also indicate that Ti-6Al-4V had significant abrasive wear loss when coupled with a ceramic counterpart. According to the mechanism, micro-abrasion plays a predominant role in the abrasion-corrosion behaviour of this material and the material losses by mechanical processes are substantially larger than losses, due to electrochemical processes.

**Keywords:** hip implant; osteolysis; tribo-corrosion; Ringer's solution

## 1. Introduction

The number of patients undergoing total hip arthroplasty is constantly rising, as is the number of patients needing a revision of the orthopedic surgery [1]. Nevertheless, patients are still encountering pain and hypersensitivity following their hip replacement. According to the UK National Joint Registry (NJR), during 2017, 91,698 primary hip and 8073 revision procedures were performed [2]. The demographic in the UK is aging, with 18% aged over 65 and 2.4% aged 85 and over [3]. The significant increase in the average age and the resultant well known age-related changes in the musculoskeletal system have led to more people presenting with the need for replacement hip implants. The trend to more sedentary lifestyles and inactivity of the population as a whole are other contributing factors to the rise in hip replacement procedures [4]. Factors which lead to revision of hip surgery include aseptic loosening, osteolysis, implant wear and fracture [3]. Mechanical wear, leading to the loosening of the implant, is one of the most frequent forms of mechanical failure [5]. Replacement materials for artificial hip joints include metals, alloys, synthetic polymers, and ceramics [6]. Increasing our understanding of tribology and corrosion behaviour in medical implants will improve the effectiveness of hips implants, and assist in the development of new joint materials.

The hip joint, and in particular, the area highlighted in Figure 1, represents a significantly challenging environment in materials science. The literature shows that the load on the hip joint can be three times the body weight normally, which can exceed ten times of the body weight during jumping. In addition, hip bones are subjected to cyclic loading as high as $10^6$ cycles in 1 year [7]. To tackle this, new bone biomaterials need to be identified with suitable mechanical properties and the necessary biocompatibility. The biocompatibility of a material describes its behaviour toward human body tissues. The wider problems posed by biocompatibility are out with the scope of this study [8].

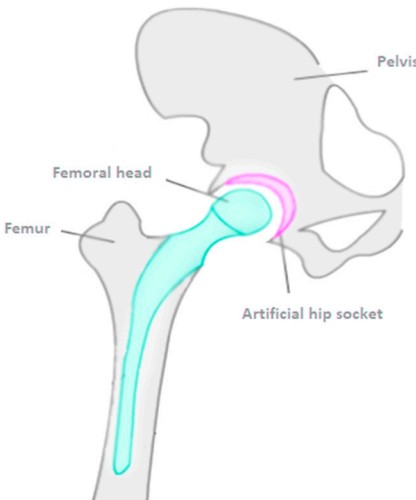

**Figure 1.** The area highlighted, susceptible to tribo-corrosion.

Many factors, including an ageing population, and a less active younger population, are generating a demand for artificial joints with better long-term performance [4]. Hip joints undergo complicated movements, including relative sliding, rotating, and flexing motions. These movements and tribo-corrosion behaviour of the implant is affected by the hip joint internal environment. A detailed understanding of these properties is therefore required to help predict the service life of artificial joints and to explore ways to enhance it. Presently, the challenge of measuring these properties is an area of intense research. Continuous development in improving joint implants and a deeper understanding of the wear-and-tear processes may contribute to future patients being given a much more positive outlook. In the UK, concerns around factors such as an ageing population and an increase in chronic diseases requiring health interventions such as hip replacements mean that there is an increasing demand and cost related to the health care service [9–14].

In this paper, micro-abrasion-corrosion test results and the analysis of a Ti-6Al-4V ELI alloy against the ceramic counterpart in a pure Ringer's solution and Ringer's solution containing SiC particles slurry are presented. The electrochemical technique was used in conjunction with the mechanical apparatus for this experimental work. Tests were run for a range of normal loads and electrical potentials to carry out potentiostatic and cyclic sweep for the investigation of micro-abrasion-corrosion behaviour of this material. The topographies of wear scars were examined by using advanced surface analysis technique, SEM. Different wear maps are presented for the interpretation of test results as a function of applied load and electrical potential by using a tribo-corrosion system approach. Wear modes, mechanisms, and synergy between mechanical and corrosion processes are identified by the construction of wastage, mechanistic and synergistic maps of this tribo-corrosion system.

## 2. Experimental

### 2.1. Test Materials

In this study, a Ti-6Al-4V ELI alloy was tested against an $Al_2O_3$ ceramic ball as a counterpart, metal-on-ceramic (MoC) coupling, in Ringer's solution, with and without SiC particles. The Titanium

material was selected due to its biocompatibility [14]. Studies have shown that it forms a protective film [15,16]. The chemical composition of the annealing medical grade Ti-6Al-4V is shown in Table 1.

**Table 1.** Chemical composition of Ti-6Al-4V.

| Element | C | Fe | O | N | H | Al | V | Ti |
|---|---|---|---|---|---|---|---|---|
| %Composition | 1 | 0.21 | 0.1 | 1 | 1 | 6.34 | 4.25 | Bal |

Conventional ceramics are used in such environments due to strength and stability in physiological conditions [17]. It is been shown that, in other studies, ceramic materials decrease osteolysis, compared to polymeric materials used as a bearing surface [18]. The mechanical properties of Ti-6Al-4V, $Al_2O_3$, and SiC abrasive particle are shown in Table 2.

**Table 2.** Mechanical and physical properties of Ti-6Al-4V, $Al_2O_3$, and SiC.

| Tribology Contact Conditions | Material | Density (kg/m$^3$) | Vickers Hardness (Hv) |
|---|---|---|---|
| Specimen | Ti 6Al-4V ELI (Grade 23) | 4500 | 341 |
| Ball | $Al_2O_3$ | 3987 | 2600 |
| Abrasive Particles | SiC | 3100 | 2700 |

The electrolyte utilized was Ringer's solution (concentration of 0.25 g·cm$^{-3}$), and is a simulated biological fluid [19]. The pH of the solution was 6.8, and remained constant for the duration of the tests. The temperature of the solution during testing was kept around 25 °C. The chemical composition of Ringer's solution is presented in Table 3.

**Table 3.** Ringer's solution composition.

| Components | Composition (g·L$^{-1}$) |
|---|---|
| NaCl | 8.6 |
| KCL | 0.30 |
| $CaCl_2$ | 0.33 |

Two test solutions were used in the series of experiments, namely a pure solution, as outlined in Table 3, and an abrasive slurry containing silicon carbide particles. The average abrasive particle size was 3 μm. The particles were mixed with the Ringer's solution to a concentration of 10 g·L$^{-1}$. The properties can be seen in Table 2.

*2.2. Experimental Procedures*

A micro-scale abrasion apparatus, in conjunction with corrosion monitoring equipment, was selected for simulating hip conditions, for several reasons. The experimental work was conducted on the apparatus, shown schematically in Figure 2; this consisted of a micro abrasion tester (Plint TE-66, Phoenix Tribology, Woking, UK) and an integrated Gill AC potentiostat (ACM Instruments, Woking, UK). The apparatus operates in accordance with British Standards EN 1071-6: 2007. A built-in counter controls the number of shaft revolutions and experimental time. The electrolyte solution was supplied to a point closely above the sample-ball contact, delivered by an integral peristaltic pump from a separate glass-container. An $Al_2O_3$ ball of 25 mm diameter was used as a counterpart, which was held between two co-axial shafts, which were rotating in unidirectional way by a variable speed DC electric motor.

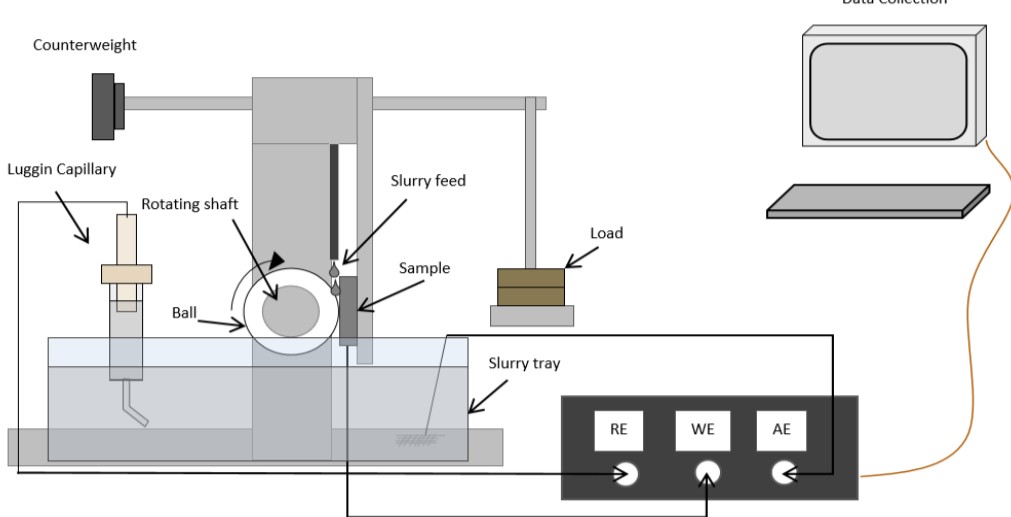

**Figure 2.** Schematic of tribo-corrosion experimental setup.

Ti-6Al-4V test samples were pre-machined from titanium wrought alloy rod stock. The samples were cut into an irregular octagon before testing. This shape maximized the number of tests that could be performed per sample, as it allowed for numerous different orientations and alignments that could be set up on the test apparatus. The samples were cut to dimensions of $28 \times 28 \times 3$ mm. The sample surface was insulated with a non-conductive tape; a surface area of 1 cm$^2$ was kept exposed for wear testing. Wear scars were generated by clamping the samples to an L-shaped arm; the arm could be rotated and the sample pressed on the ball. A range of applied loads were tested, by changing dead weights to cantilever loading arrangement. The arm, which holds the sample, could be moved horizontally to allow several tests to be carried out on a single sample. In addition, the ball was incrementally rotated in its fixings after each test, to prevent any deformation of the spherical shape of the ball.

The electrochemical interfacing consisted of a potentiostat and three electrode attachments. The working electrode (WE) was fixed to the back of the test sample, while the auxiliary electrode (AE) and reference electrode (RE) were placed in the solution bath. The reference electrode provided a stable, well defined potential, and was used to register the potential of the working electrode, i.e., the test sample. For the reference electrode, a saturated calomel electrode was used via a capillary tube, which was filled with the test solution. The auxiliary electrode was used to measure or control the current and is usually made of inert materials such as platinum, gold, or graphite [20]. A platinized- titanium (Pi-Ti) mesh formed the auxiliary electrode in this study. Sequencer software was used to carry out potentiostatic and potenti-dynamic tests. In this way, the applied voltage was monitored, polarization curves were created, and a data set was collected for the estimation of micro-abrasion-corrosion behaviour of this system.

Prior to starting a test, the surface of each sample was cleaned using deionized water, and then allowed to dry in air. The sample was also held at the electrical potential required for each test, with no normal load applied whilst submerged in solution, in order to stabilize the specimen surface voltage. For the tests, including abrasive particles, a Stuart UC151 magnetic stirrer was used to ensure a constant concentration of particles in the solution throughout the experiment.

Electrochemical Tests

Generally, the Tribo-corrosion behaviour of materials is investigated by using the electrochemistry technique. In this process, exposed surfaces of the specimens are investigated/examined prior to the test, and during test carry out. Through this process, the way in which wear influences the electrochemical response of materials exposed to the specific corrosive environment, under determined mechanical

conditions, can be evaluated. For this study, potentiodynamic and potentiostatic electrochemical testing was used to evaluate tribo-corrosion phenomena.

To study the effects of the presence of the particles in the solution, potentiodynamic tests were conducted, with and without particles. For the potentiodynamic tests, an electrical potential from −750 to +750 mV was applied. The corrosion potential $E_{corr}$, as well as the corrosion current density $I_{corr}$, were determined from the polarization curves. Moreover, 25 potentiostatic tests without abrasive particles were conducted. To provide a comparison, a further 25 potentiostatic tests were completed with abrasive particles. Changes in mass loss and the contributions by corrosion to the total mass loss were found. The performance of Ti-6Al-4V in pure abrasion conditions for each applied load was also investigated, by applying −960 mV to create cathodic conditions for these tests. For both electrochemical techniques, testing was repeated for increasing applied loads using fresh electrolyte on a fresh sample site. The exposure time was kept constant at 30 min and the speed of the ball and pump was 100rpm for all tests. A summary of the testing parameters is given in Table 4.

**Table 4.** Electrochemical test parameters.

| Applied Loads (N) | 0.5, 1, 2, 3, 4 |
|---|---|
| Applied Potentials (mV) | −960, −600, −300, 0 and +200 |
| Ball Speed (rpm) | 100 |
| Test Duration (s) | 1800 |
| Polarization Sweeps (−750 to +750 mV) with applied loads (N) | 0.5, 1, 2, 3, 4 (with and without particles) |

## 3. Results & Discussion

### 3.1. Polarisation Curves

Figures 3 and 4 are the polarization curves generated from the potentiodynamic tests for all applied loads, both without and with particles. On average, the corrosion potential ($E_{corr}$) for the tests with particles is lower than the tests without particles. This is supported by both tests at 3 N and 4 N, having values below −500 mV. This suggests that corrosion begins to take place at lower potentials when with an increase in load in the presence of abrasive particles. This could be due to abrasive particles accelerating the removal of a protective oxide layer from the surface of the Ti-6AL-4V, hence the corrosion process begins earlier, at 3 N and 4 N. Both curves suggest that the surface electrochemical activity is reduced at lower applied loads. The polarization curve at 0.5 N applied load for both conditions is observed to remain towards a corrosion potential of around −250 mV, indicating the lowest electrochemical activity. Figure 3 displays more uniformity in $E_{corr}$ values for all loads in the absence of abrasive particles. In the case of the solution without abrasives, $E_{corr}$ appears to have remained relatively unchanged over the range of applied loads, unlike the solution with particles. This indicates that the corrosion potential is more unpredictable in the presence of particles.

The corrosion current densities are relatively similar in both, suggesting that the corrosion rate is the same in the presence of abrasive particles as it is in the absence of abrasive particles. However, as the load increased for the abrasive solution, the maximum currents tended to increase. The highest current densities were recorded for 2 and 4 N, suggesting an increasing corrosion at those loads. As the applied load reached 4 N in the case of abrasives, a cathodic shift is observed. Over the spectrum of applied loads, it appears that electrochemical activity at the contact surface may be enhanced at higher loading; it is likely that a transition in lubrication mode occurs as a result of changing the contact pressure, which may explain the cathodic shift at 4 N.

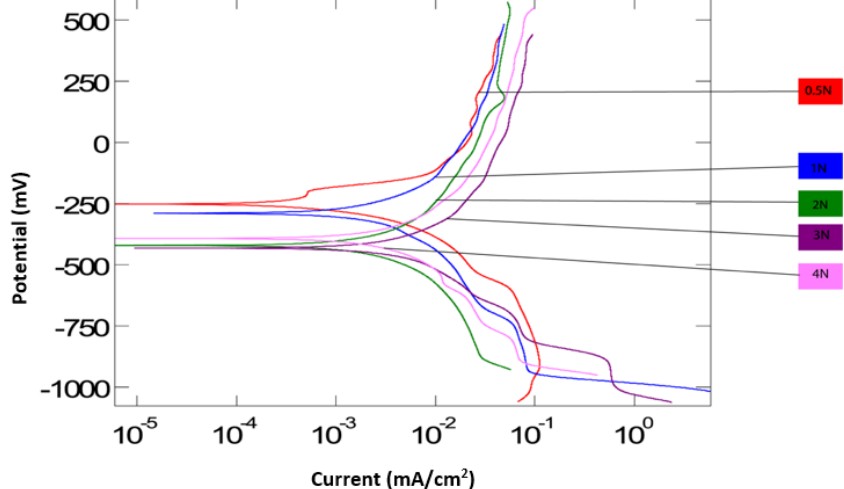

**Figure 3.** Polarization curves in pure Ringer's solution for −750 to +750 mV (1500 mV sweep).

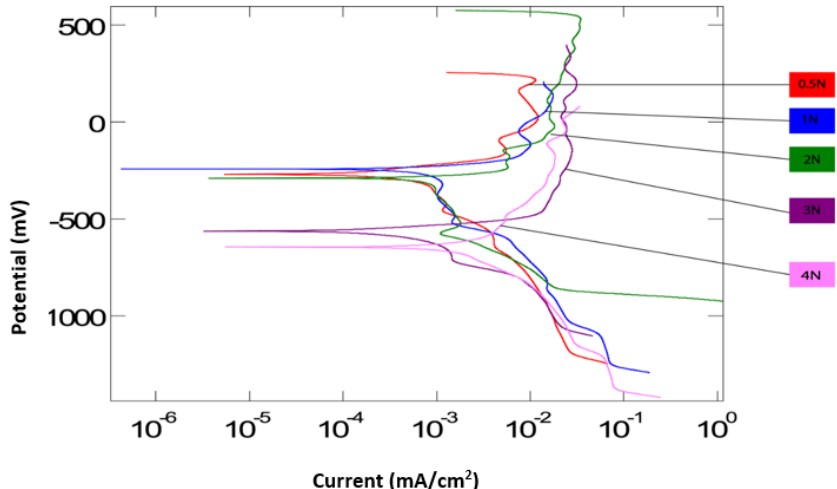

**Figure 4.** Polarization curves in pure Ringer's solution plus SiC particles slurry for −750 to +750 mV (1500 mV sweep).

*3.2. Mass Loss*

Mass loss data was produced following an established method [21,22]. The total mass of material lost due to micro-abrasion-corrosion, $K_{ac}$ (g), is equal to the sum of mass lost due to micro-abrasion, $K_a$ (g), and the mass loss due to corrosion, $K_c$ (g).

$$K_{ac} = K_a + K_c \tag{1}$$

The total mass loss ($K_{ac}$) was calculated by multiplying the volume of the wear scar ($V$) by the density of the material. The mass loss volumes were estimated using a standard technique for measuring wear scar of spherical geometry. It is assumed that the shape of the crater conforms to the shape of the ball [23]. Wear volume ($V$) may be calculated by the measurement of the crater diameter ($b$) and $R$ the ball radius, as shown below:

$$V = \frac{\pi b^4}{64R} \, (For \, b \ll R) \tag{2}$$

Mass loss due to corrosion ($K_c$) was estimated using Faraday's Law, as follows:

$$K_c = \frac{Q}{ZF} \tag{3}$$

Equation (3) may also be written as:

$$K_c = \frac{MI_{corr}t}{ZF} \tag{4}$$

where $Q$ is the charge passed (C); M is the atomic mass of the test material, $I_{corr}$ is the corrosion current density (mA/cm$^2$), $t$ is the test duration, $Z$ is the number of valence electrons in the test and $F$ is Faraday's constant (96,500 C·mol$^{-1}$).

Micro-abrasion weight loss ($K_a$) can be further divided up into pure micro-abrasion weight loss ($K_{ao}$) and the synergistic effect of corrosion on the micro-abrasion ($\Delta K_a$):

$$K_a = K_{ao} + \Delta K_a \tag{5}$$

The pure micro-abrasion mass loss ($K_{ao}$) was calculated using Equation (2); the material density and the wear scars from the pure micro-abrasion tests (cathodic conditions at $-960$ mV) for each applied load. Corrosion mass loss ($K_c$) can similarly be divided into pure corrosion mass loss ($K_{co}$) and the additive effect of micro-abrasion on the corrosive mass loss ($\Delta K_c$):

$$K_c = K_{co} + \Delta K_c \tag{6}$$

Approximate values of pure corrosion weight loss ($K_{co}$) were calculated using Equation (3) and the current density values from the polarization curves with and without particles for every applied load.

When the synergy components have a negative value, it is termed "antagonistic", as the two wear mechanisms work against each other instead of together. Therefore, the total micro-abrasion-corrosion mass loss can be represented by the following equation:

$$K_{ac} = K_{ao} + \Delta K_a + K_{co} + \Delta K_c \tag{7}$$

The mass loss results using this analysis are shown in Tables 5 and 6. Total mass losses, with and without particles in Ringer's solution, are shown in Figure 5.

**Table 5.** Micro-abrasion-corrosion mass loss in pure Ringer's solution.

| Applied Potential (mV) | Applied Load (N) | $K_{ac}$ (µg) | $K_c$ (µg) | $K_a$ (µg) | $K_c/K_a$ |
|---|---|---|---|---|---|
| −600 | 0.5 | 193.89 | 3.76 | 190.13 | 0.0198 |
| | 1 | 350.27 | 0.5 | 349.77 | 0.0014 |
| | 2 | 782.98 | 0.69 | 782.29 | 0.0009 |
| | 3 | 770.91 | 2.57 | 768.34 | 0.0033 |
| | 4 | 1165.87 | 5.54 | 160.33 | 0.0346 |
| −300 | 0.5 | 245.19 | 9.51 | 235.68 | 0.0404 |
| | 1 | 477.54 | 6.14 | 471.40 | 0.0130 |
| | 2 | 911.62 | 1.78 | 909.84 | 0.0020 |
| | 3 | 925.30 | 4.36 | 920.94 | 0.0047 |
| | 4 | 1182.32 | 7.13 | 1175.19 | 0.0061 |
| 0 | 0.5 | 216.09 | 9.31 | 206.78 | 0.0450 |
| | 1 | 377.60 | 4.75 | 372.84 | 0.0127 |
| | 2 | 690.29 | 4.36 | 685.93 | 0.0064 |
| | 3 | 1010.66 | 0.73 | 1009.92 | 0.0007 |
| | 4 | 1284.71 | 7.33 | 1277.38 | 0.0057 |
| +200 | 0.5 | 124.74 | 3.96 | 120.78 | 0.0328 |
| | 1 | 399.12 | 7.72 | 391.39 | 0.0197 |
| | 2 | 636.70 | 9.71 | 626.99 | 0.0155 |
| | 3 | 820.04 | 1.58 | 818.46 | 0.0019 |
| | 4 | 1302.01 | 7.33 | 1294.68 | 0.0057 |

**Table 6.** Micro-abrasion-corrosion mass loss in pure Ringer's solution plus SiC particles slurry.

| Applied Potential (mV) | Applied Load (N) | $K_{ac}$ (µg) | $K_c$ (µg) | $K_a$ (µg) | $K_c/K_a$ |
|---|---|---|---|---|---|
| −600 | 0.5 | 10.46 | 4.91 | 5.55 | 0.8846 |
| | 1 | 35.02 | 6.93 | 28.09 | 0.2467 |
| | 2 | 47.75 | 6.16 | 41.59 | 0.1481 |
| | 3 | 37.76 | 4.79 | 32.97 | 0.1453 |
| | 4 | 39.91 | 7.73 | 32.24 | 0.2345 |
| −300 | 0.5 | 64.72 | 5.49 | 59.23 | 0.0927 |
| | 1 | 78.30 | 0.00 | 78.30 | 0.00 |
| | 2 | 91.16 | 1.71 | 89.45 | 0.0191 |
| | 3 | 69.03 | 0.48 | 68.55 | 0.0070 |
| | 4 | 63.67 | 9.74 | 53.93 | 0.1806 |
| 0 | 0.5 | 91.16 | 0.69 | 90.47 | 0.0099 |
| | 1 | 77.09 | 2.65 | 74.44 | 0.0356 |
| | 2 | 91.16 | 4.44 | 86.72 | 0.0512 |
| | 3 | 101.07 | 0.73 | 100.34 | 0.0073 |
| | 4 | 82.00 | 1.66 | 80.34 | 0.0207 |
| +200 | 0.5 | 128.47 | 3.76 | 124.41 | 0.0302 |
| | 1 | 116.59 | 5.59 | 111.00 | 0.0504 |
| | 2 | 118.23 | 7.15 | 111.08 | 0.0644 |
| | 3 | 128.47 | 7.41 | 121.06 | 0.0612 |
| | 4 | 130.24 | 7.25 | 122.99 | 0.0589 |

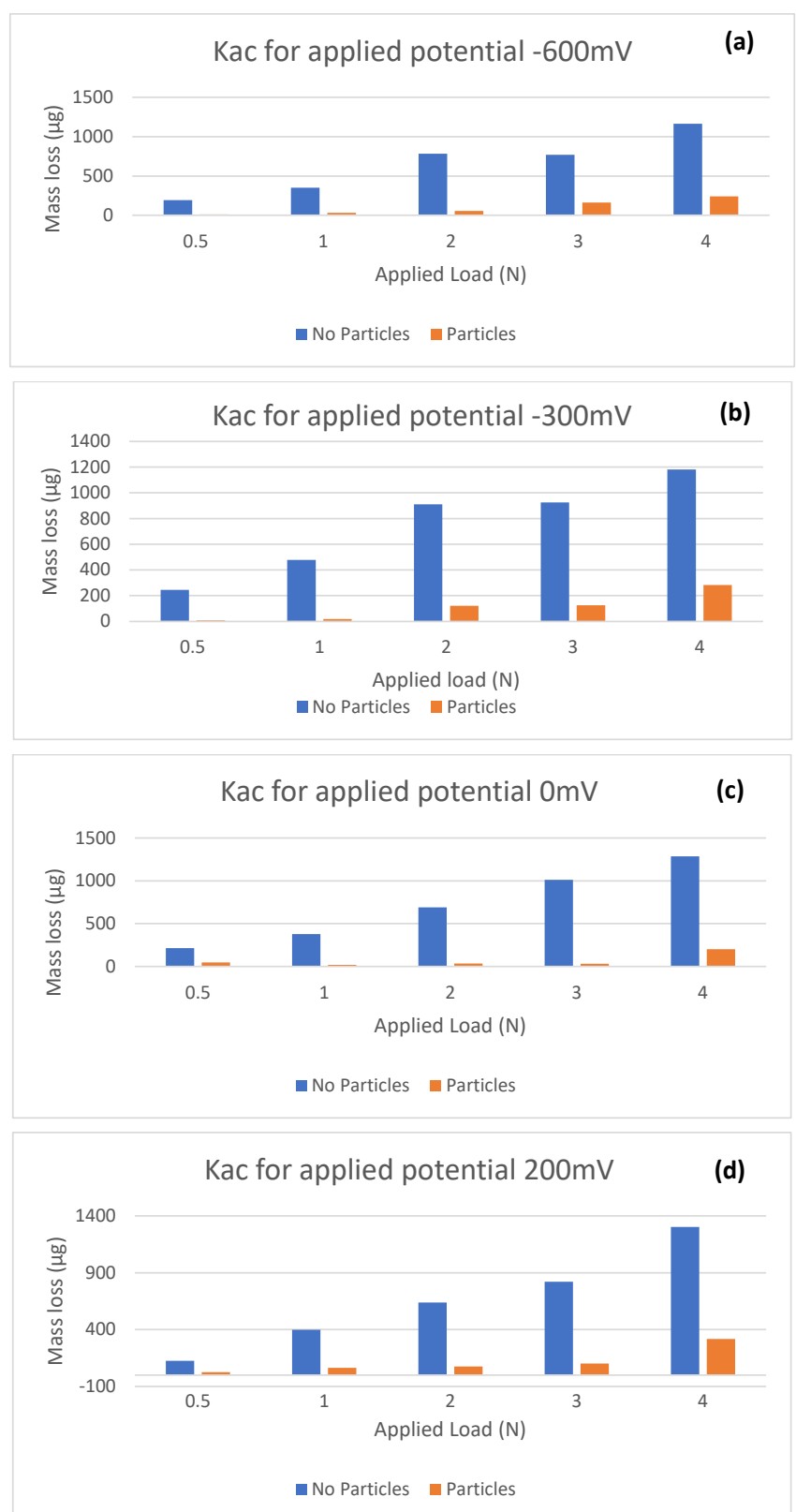

**Figure 5.** Micro-abrasion-corrosion mass loss comparison in pure Ringer's solution, versus pure Ringer's solution plus SiC particles slurry, (**a**) at −600 mV, (**b**) at −300 mV, (**c**) at 0 mV, and (**d**) at 200 mV.

Tables 5 and 6 show that, for every test condition, the total and micro-abrasion mass loss values are very higher in magnitude for tests without solid particles, whilst the corrosion mass loss is significantly

lower for both, without and with solid particles. The corrosion contribution is small, but has been shown to play a critical role in the wear synergism. It can be seen clearly that the electrochemical mass loss ($K_c$) remains a consistent fraction of the total mass loss contribution ($K_{ac}$). This is consistent with previous papers [12,20,24] and should be expected, since Ti-6Al-4V is a highly corrosion resistant material. The majority of $K_c/K_a$ values are less than 0.1, which indicates that micro-abrasion is the dominant wear regime in both conditions. At an applied potential of −600 mV with particles present, there appears to be a relationship between the mass loss due to abrasion, and mass loss due to corrosion. As the rate of micro-abrasion increases, the rate of corrosion also appears to intensify. This would indicate that, as the rate of micro-abrasion increases, the rate at which a protective passive layer is removed also increases, thus allowing for increased corrosive wear. However, this trend is not visible across all potentials, suggesting that the material is perhaps more vulnerable to corrosive wear at this specific corrosion potential.

The highest recorded value for corrosion occurs with particles present, indicating that mechanical abrasion as a result of the presence of particles may disrupt and remove this film, leading to increased corrosion. The values for abrasive wear loss are considerably higher for this study; approximately ten times more than previously recorded for Ti-6Al-4V in a bio-simulated environment [13,25]. This is likely due to the ceramic counterpart. The $Al_2O_3$ ball has a significantly higher Vickers hardness value in comparison to the polyethene counterpart (UHMWPE) used in both these previous studies. This can be assumed that abrasive wear could be accelerated in a ceramic/metal pairing, potentially reducing the hip implant lifetime. However, in this case, tests results show a different scenario, i.e., micro-abrasion wear significantly reduced in the presence of abrasive Tables 4 and 5.

According to Figure 5, the mass loss ($K_{ac}$) is considerably higher for the solution without abrasives across all applied potentials. This suggests that the two-body abrasion is the dominant wear mode in the case without particles. This reinforces that two-body abrasion is the more severe mode of wear. In the presence of SiC, hard solid particles were embedded in the contact area of the specimens to form a composite layer, thus reducing the total mass loss. It can also be seen that mass losses in both solutions tended to increase as applied load was increased; consistent with Archard's wear model, which states that the wear rate of the material is proportional to the applied load and sliding distance.

### 3.3. SEM Images of Wear Scars

Figures 6 and 7 allow for a comparison of different applied loads, with and without abrasive particles. In Figure 6a,b, there is clear evidence of directional grooving, indicating a uniform dominant two-body abrasion. However, in the presence of particles Figure 7a,b, there is no evidence of this phenomenon. Wear scars are showing embedded abrasive particles and plastic deformation, suggesting the formation of a composite layer, which results in a reduction in abrasive mass loss due to the presence of solid particles wear scars. This is consistent with the conclusions reached from the mass loss data in Figure 5. The diameter is also almost double in the absence of abrasive particles, indicating that two-body abrasive wear is more severe than that in the presence of solid particles.

Figure 6c,d show how the change in applied load can affect the type of wear that is produced and how, at some intermediate load, a transition must take place between the wear mechanisms. In these figures, the exaggeration of the parallel grooves as the applied load is increased is visible. This suggests a transition to a two-body dominated regime as the particles become entrained in the cratering ball, resulting in a grooving. Additionally, the reduction of surface ridges could contribute to this transition. In Figure 7a–d, fine SiC particles appear embedded across wear surface, including larger agglomerated particles, signifying that a deposition effect occurs across a range of applied loads. An increased number of agglomerated particles are present at the higher load, suggesting that there is a positive relationship between organic deposition and applied load. From these microscopy results, it can also be noted that the substrate material did not exhibit any form of uninform corrosion or pitting, again demonstrating the corrosion resistance of grade 23 Ti-6Al-4V alloy.

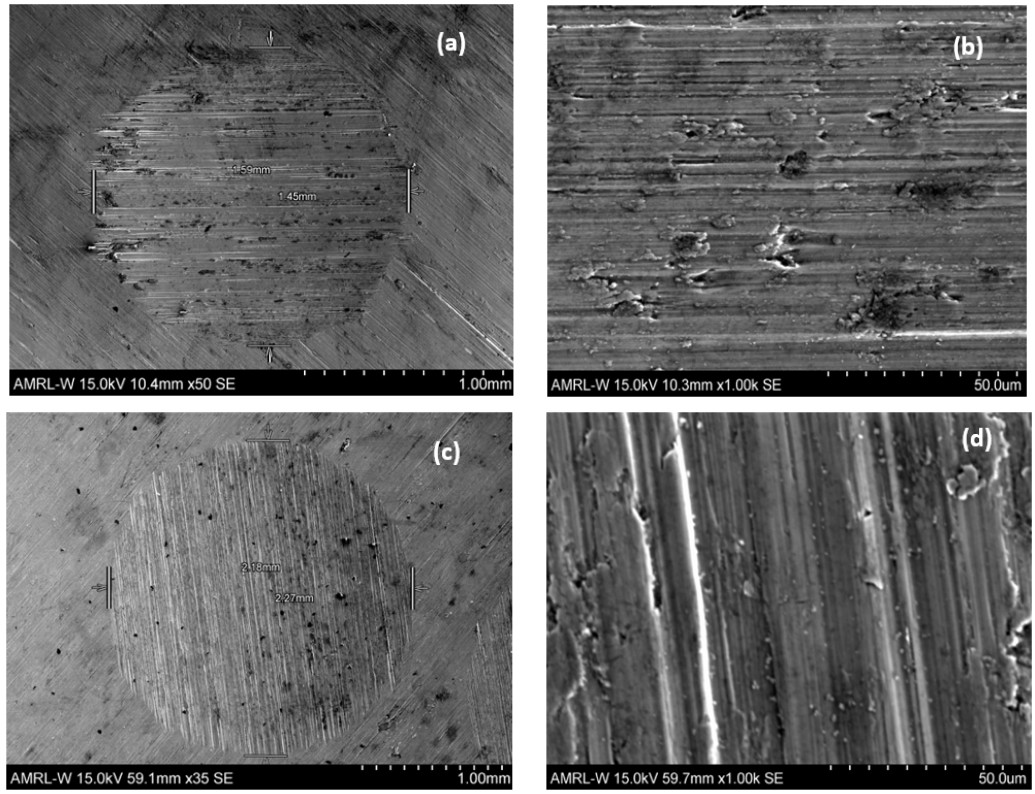

**Figure 6.** SEM images of the wear scars in Ringer's solution, (**a**) at 0.5 N without abrasive particles (**b**) 'a' at ×1000 magnification, (**c**) at 4 N without abrasive particles (**d**) 'c' at ×1000 magnification.

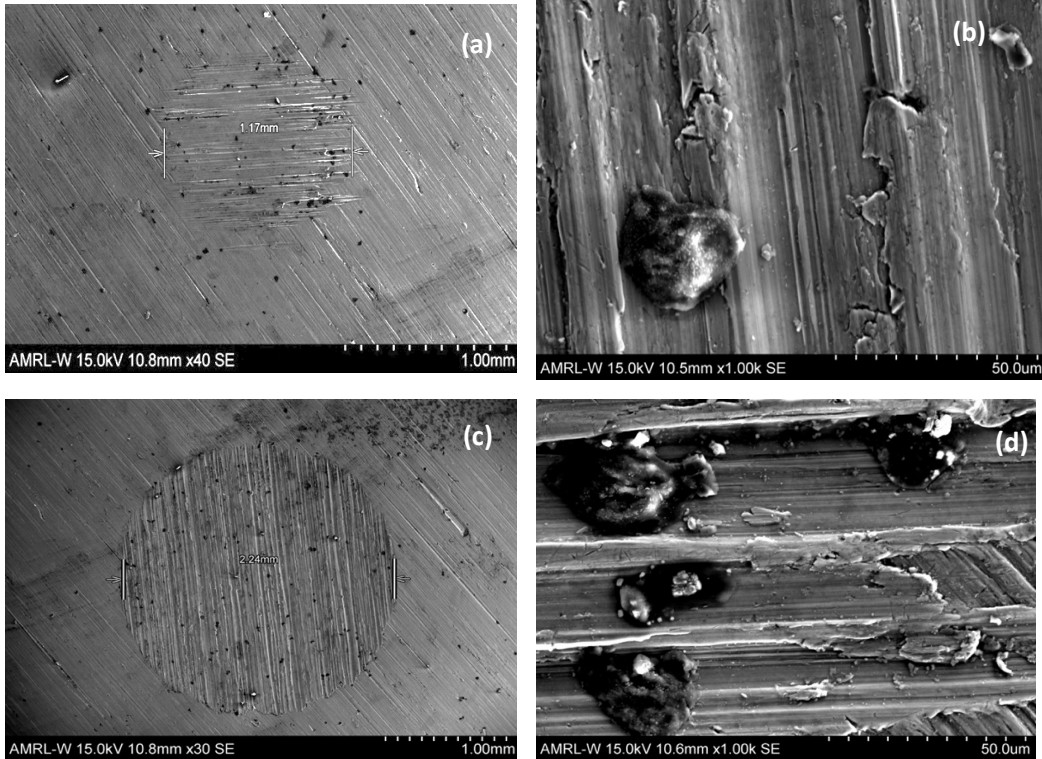

**Figure 7.** SEM images of the wear scars in Ringer's solution, (**a**) at 0.5 N with abrasive particles (**b**) 'a' at ×1000 magnification, (**c**) 4 N with abrasive particles (**d**) 'c' at ×1000 magnification.

### 3.4. Wear Maps

A series of tribo-corrosion maps have been constructed for Ti-6Al-4V in Ringer's solution. Three categories of maps were constructed: wastage, mechanism and synergy maps. The maps were drawn by plotting the results from potentiostatic tests (in the presence and absence of abrasive particles) on a chart and interpolating between the points to determine the boundary lines. It was assumed that the wear results varied linearly between each condition. Previously developed mapping techniques and their defined categories [20,24,26] were adapted as follows (Table 7):

The wastage map in the absence of abrasive particles Figure 8a demonstrates that the applied electrical potential clearly has no significant effect on the total wear compared to the applied load. This is due to the fact that the amount of material loss caused by corrosion was much lower than micro-abrasion.

The wastage map in the presence of abrasive particles Figure 8b shows that, over the cathodic range of potentials, the wastage regimes remain low despite increasing load. In the cathodic region at loads below 1 N, no significant corrosive effects appear to take place. In comparison to an earlier study [12], there is some similarity, where a high wastage regime dominates for increasing loads with anodic potentials. The map demonstrates that the low-medium wastage regime is dominant. As a result, any variation in material loss due to corrosion had no significant effect on total wear in both cases (with and without particles). This is further evidenced by the highest wastage occurring at 4 N and the lowest wastage occurring at a load of 0.5 N Figure 5.

The mechanism map of Ti-6Al-4V without particles, Figure 8c, is entirely micro-abrasion dominated, reinforcing the titanium alloys strong corrosion resistance. The mechanism map in the presence of particles Figure 8d is also micro-abrasion dominated, but displays different behaviour at cathodic potentials. At an electrical potential of -600 mV and for the range of applied loads, a micro-abrasion-corrosion region is present. This could be due to a deposition effect on the surface, as the system is cathodic, and the fact that oxide reduction occurred at low electrical potentials and low applied loads.

Moreover, from the results in Table 6, at −600 mV, despite the highest ratio of corrosion to micro-abrasion at 0.5 N, micro abrasion is still significantly greater than corrosion for the range of applied loads. The presence of a micro-abrasion corrosion regime at lower potentials is inconsistent with previous papers [12]. During tests, a constant stream of solution could not always be achieved and may have been a source of inconsistency among the tests, as the variation in forces on the sample could have affected the wear mechanisms.

In the non-abrasive synergy map Figure 8e generally, as the applied load is increased, the wear tends to change from a synergistic regime to an antagonistic regime. This could be the result of oxide layers forming a protective barrier to wear, which is then mechanically removed at higher loads, which has been observed in previous corrosion studies [21,25,27]. The synergy map for the abrasive solution Figure 8f has a significant antagonistic coverage in comparison to Figure 8e, where abrasive particles are not present. Possible reasons for this could be increased solution viscosity, improving the lubrication regime and reducing the effect of corrosion–wear due to additional particles. In addition, hard SiC particles are likely to have been embedded on the surface of the softer specimen; therefore, underlying a transition to a two-body abrasion regime towards the center of the contact zone [21]. In Tribo-Corrosion, when the wear mechanism is in the antagonistic wear regime, i.e., micro-abrasion and corrosion in this case, work against each other, instead of together. Synergistic maps, Figure 8e,f, suggest that the antagonistic behaviour of micro-abrasion and corrosion is dominated in both cases, i.e., with and without abrasive particles.

**Table 7.** Micro-abrasion-corrosion maps boundaries.

| Map Category | | Region | Boundaries |
|---|---|---|---|
| Wastage | 🟩 | Very Low | $K_{ac} \leq 0.15 * K_{ac\ max}$ |
| | 🟨 | Low | $0.15 * K_{ac\ max} < K_{ac} \leq 0.35 * K_{ac\ max}$ |
| | 🟧 | Medium | $0.35 * K_{ac\ max} < K_{ac} \leq 0.80 * K_{ac\ max}$ |
| | 🟥 | High | $0.80 * K_{ac\ max} < K_{ac}$ |
| Mechanism | 🟦 | Pure micro-abrasion | $K_c/K_a \leq 0$ |
| | 🟩 | Micro-abrasion | $0 \leq K_c/K_a < 0.1$ |
| | 🟨 | Micro-abrasion-corrosion | $0.1 \leq K_c/K_a < 1$ |
| | 🟥 | Corrosion-micro-abrasion | $1 \leq K_c/K_a < 10$ |
| | ⬛ | Corrosion | $10 \leq K_c/K_a$ |
| Synergy | 🟦 | Antagonistic | $\Delta K_a/\Delta K_c < 1$ |
| | 🟪 | Additive | $\Delta K_a/\Delta K_c \leq 0.1$ |
| | 🟧 | Additive-synergistic | $0.1 < \Delta K_a/\Delta K_c \leq 1$ |
| | 🟥 | Synergistic | $\Delta K_a/\Delta K_c > 1$ |

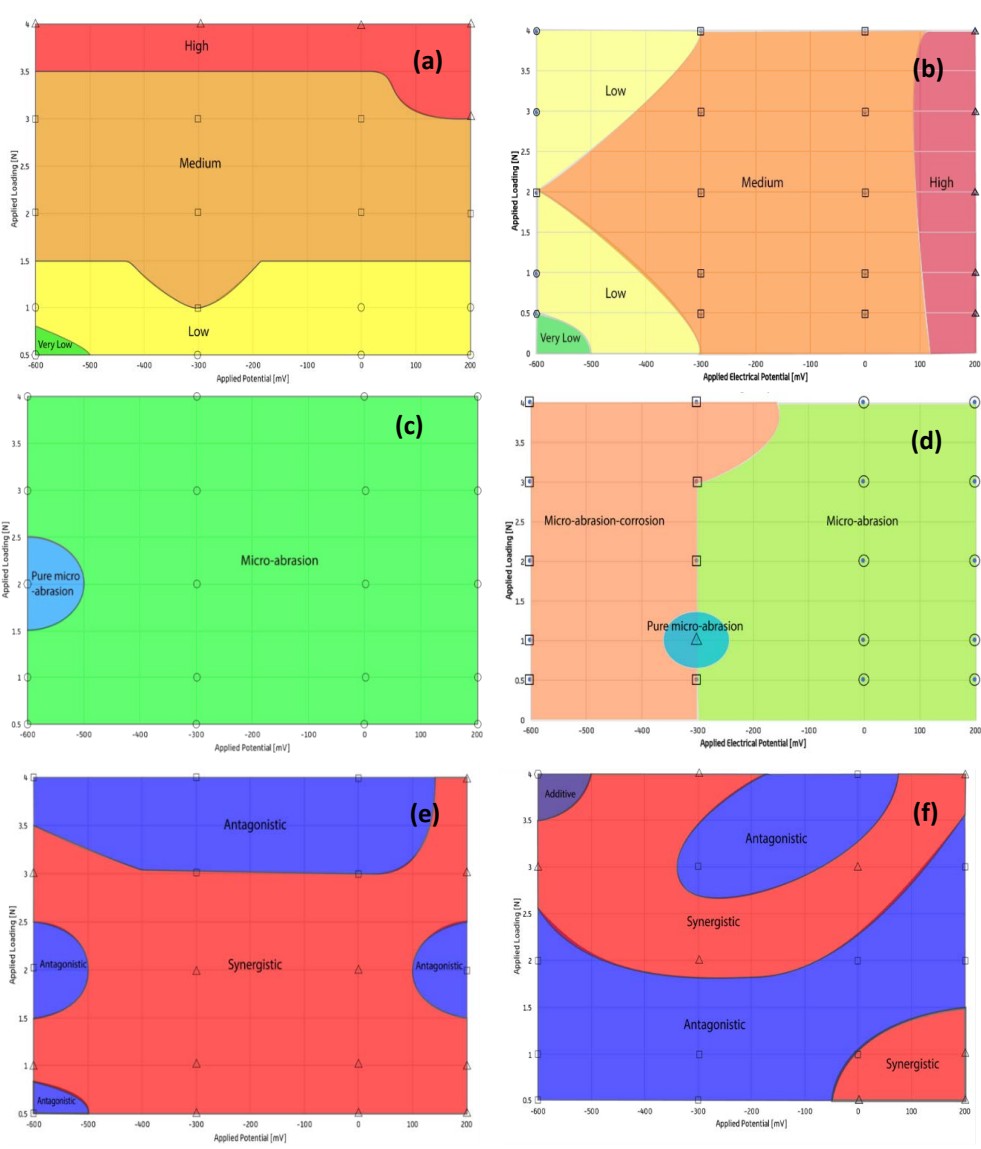

**Figure 8.** Wear maps corresponding to pure Ringer's solution Versus Ringer's solution plus SiC abrasive particles, (**a**) wastage map for pure Ringer's solution, (**b**) for pure Ringer's solution plus SiC particles, (**c**) mechanism map for pure Ringer's solution, (**d**) for pure Ringer's solution plus SiC particles, (**e**) synergy map for pure Ringer's solution, and (**f**) synergy map for pure Ringer's solution plus SiC particles.

## 4. Conclusions

- The effects of applied load and electrical potential on the micro-abrasion-corrosion of Ti-6Al-4V in Ringer's solution (with and without abrasive particles) were assessed by electrochemical techniques.
- The results from the micro-abrasion-corrosion studies were used to generate wastage, mechanism and synergy maps, to help define the material Tribo-Corrosion behaviour in simulated hip joint conditions.
- The potentiodynamic polarization curves indicated that the corrosion potential for the tests with particles was lower than the tests without particles. This could be due to abrasive particles accelerating the removal of a protective oxide layer from the surface of the Ti-6AL-4V, suggesting that, in an abrasive slurry, the corrosion process begins at lower potentials.
- In the absence of solid particles, the values for abrasive wear loss are considerably higher for this research in comparison to similar Ti-6Al-4V studies. This was likely due to the hardness value for the ceramic counterpart. A significant reduction in abrasive wear in the presence of solid particles suggests that more research work is required on a ceramic/metal pairing.
- Mechanism maps developed indicated micro-abrasion mechanisms dominated for all test conditions, confirming the corrosion resistant nature of Ti-6Al-4V. The material losses by mechanical process are appreciably larger than losses due to electrochemical processes.
- The synergy map in the presence of abrasive particles displayed significant antagonistic behavior. This was associated with the increased solution viscosity, improving the lubrication regime and therefore reducing the effect of corrosion–wear due to additional particles.
- Scanning electron microscopy identified that the presence of particles altered the dominant wear mechanism. At low applied loads, two body wear was dominant in the absence of particles, but in the presence of abrasive particles 3, plastic deformation was more prevalent, with evidence of 3 body wear. At higher loads in the presence of particles, there was a transition to a two-body dominated regime.

**Author Contributions:** Conceptualization, M.M.S. and G.R.; methodology, Y.E.S.; validation, Y.E.S.; formal analysis, Y.E.S.; investigation, Y.E.S., resources, M.M.S.; data curation Y.E.S.; writing—original draft preparation, Y.E.S.; writing—review and editing, G.R. and M.M.S.; visualization, supervision, G.R. and M.M.S.; project administration, G.R. and M.M.S. All authors have read and agreed to the published version of the manuscript.

**Funding:** This research received no external funding.

**Conflicts of Interest:** The authors declare no conflict of interest.

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
