# Peer review of "Mapping Tribo-Corrosion Behaviour of TI-6AL-4V Eli in Laboratory Simulated Hip Joint Environments"

_lubricants, doi:10.3390/lubricants8070069_

Round 1

Reviewer 1 Report

The article is entitled: ‘Mapping tribo-corrosion behavior of Ti-6Al-4V Eli in laboratory simulated hip environments’. The authors did present the map approach to describe some synergy, micro abrasion, etc.

The experimental approach sounds well managed. On Figure 2, AE is on the scheme but not in legend. One may think about CE. A lot of mismatch is pointed out through the text, see detailed comments. The authors should pay attention on the physical unit mV instead of mv through the text, thanks.

It should be detailed some outlooks related to this kind of investigations, detailing some experiments that should be investigated. The antagonistic concept, according to my opinion, needs some improvements through the definition and or adding some references.

Author Response

Dear Reviewer,

Many thanks for your excellent review, which has enhanced my knowledge in this field.

Reviewer 2 Report

The paper presents the tribo-corrosion behaviours of Ti-6Al-4V ELI, which is important for application in artificial hip replacements. Although some interesting results were provided, more information should be clearly stated, so the manuscript needs to be revised. My comments and concerns are listed below:

  1. Ringer's solution, the R should be capitalised throughout the manuscript.
  2. What are the contact pressure and temperature of the electrochemical wear testing? I think these parameters are important to simulate a hip joint environment.
  3. Authors describe (CE) counter electrode in text but it was labelled as AE in Figure 2.
  4. Spelling error on line 65, “Tests were run for a rage of normal loads…”
  5. Typing error on line 287, “ In in these figures,”.
  6. Please describe Figure 8 (b) in text.
  7. On line 356, “ Moreover, from the results table 6, the highest ratio of corrosion to micro-abrasion (0.5 at -600 mv), micro abrasion is still three times greater than corrosion.” This sentence is not clear, please reiterate.
  8. Can authors provide a suggestion on how is the study of tribo-corrosion of Ti-6Al-4V ELI in the presence of SiC particles and without helps in artificial hip replacements? Which experimental setup (with or without SiC particles) is more corrosion resistant and wear resistant?

Author Response

(The authors gave the same response as above.)

Reviewer 3 Report

I have read with a great interest your paper titled "Mapping tribo-corrosion behaviour of TI-6AL-4V Eli in laboratory simulated hip environments". The topic is interesting and worth investigating.  The methodology used to investigate the tribocorrosive properties of the Ti-6AL-4V and used for discussion are also reasonable and clear.The paper is suggested to be accepted after following issues are addressed. I would like to share with you my main comments:

  1. Page1 line31-32 "...procedures were performed [2]", adding a period
  2. Page 2 line 46-47  "....of the body weight during jumping.." delete a period
  3. Page 2 line 49 "The biocompatibility a material describes....." ---> "The biocompatibility of a material describes....."
  4. Page 2 line 63 " .. and ringer solution SiC particles..."  ----> "and ringer solution containing SiC particles"
  5. Page 3 line 82-83 " The mechanical properties Ti-6Al-4V," --->" The mechanical properties of Ti-6Al-4V,"
  6. Page 12 line 33 "and their defined categories [2024,26]” ----> "and their defined categories [20, 24,26]"
  7. Table 7 is suggested to redrawn.

Author Response

(The authors gave the same response as above.)
